# MaterialSeg3D: Segmenting Dense Materials from 2D Priors for 3D Assets

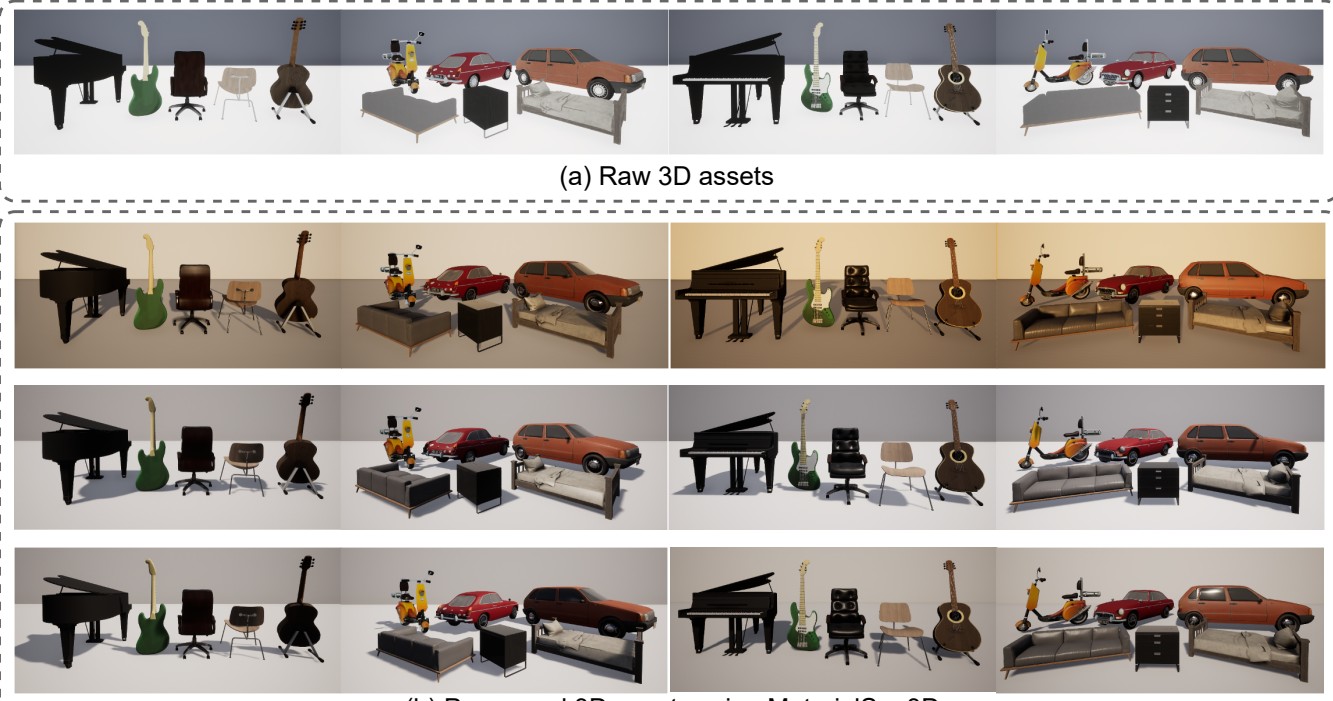

(a) Raw 3D assets

(b) Processed 3D assets using MaterialSeg3D

**Figure 1: (a) Renderings of raw 3D assets that only have albedo information. (b) Renderings of processed assets with material information under different lighting conditions. Given a raw asset, our MaterialSeg3D can actively predict and refine dense explicit surface material based on 2D priors. Equipped with material definitions, 3D assets support physically based rendering, leading to photorealistic visual effects.**

## ABSTRACT

Driven by powerful image diffusion models, recent research has achieved the automatic creation of 3D objects from textual or visual guidance. By performing score distillation sampling (SDS) iteratively across different views, these methods succeed in lifting 2D generative prior to the 3D space. However, such a 2D generative image prior bakes the effect of illumination and shadow into the texture. As a result, material maps optimized by SDS inevitably involve spurious correlated components. The absence of precise material definition makes it infeasible to relight the generated assets reasonably in novel scenes, which limits their application in downstream scenarios. In contrast, humans can effortlessly circumvent this ambiguity by deducing the material of the object from its appearance and semantics. Motivated by this insight, we propose *MaterialSeg3D*, a 3D asset material generation framework to infer underlying material from the 2D semantic prior. Based on such a prior model, we devise a mechanism to parse material in 3D space. We maintain a UV stack, each map of which is unprojected from a specific viewpoint. After traversing all viewpoints, we fuse the stack through a weighted voting scheme and then employ region unification to ensure the coherence of the object parts. To fuel the learning of semantics prior, we collect a material dataset, named *Materialized Individual Objects (MIO)*, which features abundant images, diverse categories, and accurate annotations. Extensive quantitative and qualitative experiments demonstrate the effectiveness of our method.

## CCS CONCEPTS

• **Computing methodologies** → **Image processing**; **Texturing**; **Mesh geometry models**.

*ACM MM, 2024, Melbourne, Australia*
© 2024 Copyright held by the owner/author(s). Publication rights licensed to ACM.
ACM ISBN 978-x-xxxx-xxxx-x/YY/MM
https://doi.org/10.1145/nnnnnnn.nnnnnnn

**Unpublished working draft. Not for distribution.**

# KEYWORDS

Image-based Rendering, Texture Mapping, Texture Synthesis

## 1 INTRODUCTION

3D asset creation, as a pivotal topic in computer graphics, has great application potential in virtual reality, augmented reality, games, and movies. It is a laborious workload for the artist in the traditional industrial pipeline. To create a 3D object of high quality, the artist often spends several days on sculpting geometry and drawing texture. The creation should adhere to some commonly recognized principles, such as neat polygon mesh and proper material design. This paper focuses on the material assignment of 3D assets. We follow Disney-principled BRDF and employ roughness and metallic as the primary physical properties of the material. These properties modulate the BRDF terms in the rendering equation and enable realistic re-lighting effects in different illumination conditions. With the advance of generative modeling, recent research [13, 39, 71] has achieved automatic creation of 3D objects according to textual or visual description. Most current methods resort to powerful 2D generative image models to supervise the 3D content generation. However, such 2D supervision bakes illumination. In this case, score distillation sampling inevitably leads to entangled material maps. Without precise material information, the generated assets cannot be re-lit realistically in novel scenes, which limits their application scope greatly.

For better usability, it is desirable to generate Physically-Based Rendering (PBR) material maps during asset creation [58]. We first investigate how the artist completes such a challenge. Given reference images of the object-of-interest, the artist can infer the material properties of each part according to the semantic information and appearance. For example, assuming an armchair with silver legs, thick black cushions, and a backrest, a human can confidently determine that the legs are metal and the seat cushion might be leather. Inspired by such a phenomenon, we point out that 2D priors' knowledge of material information can serve as powerful guidance for 3D material. Intuitively, material segmentation on 2D images is a perception-based method that can distill knowledge from labeled training images. However, existing material-related segmentation datasets such as DMS [65] or MINC [7] only provide material labels for open scenes including multiple instances, which are less reliable in dealing with single-object component segmentation. With the motivation of establishing a database to construct 2D material prior knowledge for individual objects, we collect *Materialized Individual Objects (MIO)*, a novel 2D single-object segmentation dataset consisting of dense material semantic annotations of objects with intricate semantic classes and captured camera angles. Images are **(a)** collected from both real-world captures and 3D asset renderings, augmenting the prior knowledge from reality and easing the domain gap; **(b)** sampled with various camera angles including but not limited to top and side views; **(c)** annotated and supervised by professional annotators. For each material class label in the dataset, we assign PBR material (Metallic, Roughness) under instructions of prior knowledge from experienced modelers. The MIO dataset contributes to establishing robust prior knowledge in material information while narrowing the distribution gap between object renderings in the application and the training data as well.

Empowered by the MIO dataset, we manage to propose *MaterialSeg3D*, a workflow that can automatically predict and generate precise surface material for 3D objects. Taking the geometry mesh and Albedo UV of an asset as input, our method first renders multi-view images of the asset with a manually and randomly selected camera pose. These multi-view renderings are then inferred by the material segmentation model, which is trained beforehand on the MIO dataset. Each predicted material result of multi-view images is further projected back onto a temporary UV map with the corresponding camera matrix. The final UV map for material labels is calculated through the voting mechanism and is further converted into a PBR material UV map including the Metallic and Roughness score for each material label assigned in the MIO dataset. As shown in Fig. 1, by absorbing 2D prior knowledge of material information from the MIO dataset, MaterialSeg3D can generate accurate surface material for 3D assets, resulting in vivid rendered visuals and application potential in the real world.

To summarize, the contributions of this paper are:

- We innovatively propose to utilize human prior knowledge of 2D material information in the surface material generation of 3D assets. Prior knowledge of the inherent relationship between the semantics and materials offers more reliable and precise guidance.
- We construct *MIO* dataset, which is currently the largest multiple-class single asset 2D material semantic segmentation dataset including images captured from especial camera angles and patterns, and each image is accurately annotated by a professional team.
- We introduce *MaterialSeg3D*, a novel workflow that can infer underlying material from the 2D semantic prior and accurately generate precise surface material for different parts of the 3D asset. This method can be significant in improving the quality of 3D assets from existing open-source datasets or websites.

## 2 RELATED WORK

### 2.1 3D Asset Generation

Early methods in 3D asset generation often adapted existing 2D convolutional neural networks (CNNs) and generative adversarial networks (GANs) to generate 3D voxel grids [24, 30, 45, 61, 69, 74], these methods are straightforward but also difficult to generate high-quality 3D assets because they have many limitations such as high memory usage and computational complexity.

Subsequent research explored more methods such as based on point clouds [2, 46, 50, 76, 81], and implicit functions [14, 48]. The biggest problem of these 3D characterizations is the lack of compatible performance on standard computer graphics. Then to improve the quality and efficiency of 3D asset generation, the mesh-based 3D generative models [26, 29, 38, 42, 56, 64, 80] have emerged, accommodating complex topologies and shapes with varying resolutions. Importantly, the results from these models can seamlessly integrate with standard graphics engines, aligning with current industry demands for effective 3D data representation.

Currently, mainstream 3D generative models largely rely on text guidance to create a variety of 3D assets. Some methods involved optimizing Neural Radiance Fields (NeRF) [49] through text-image

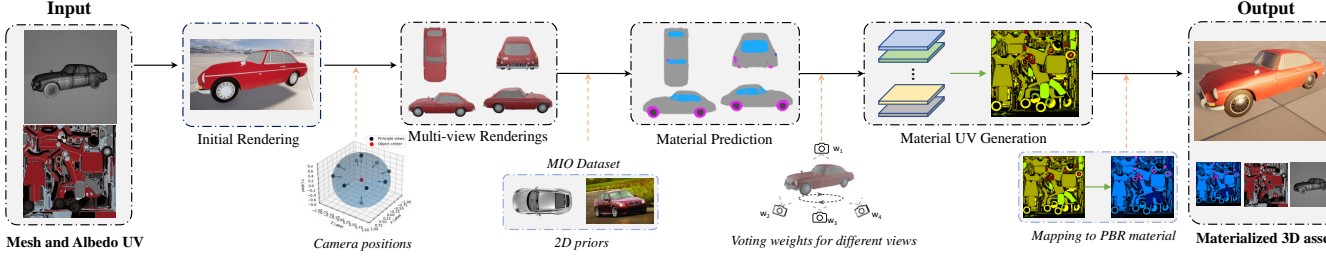

**Figure 2: Overall framework of our MaterialSeg3D workflow. The material segmentation model is trained on MIO beforehand. Multi-view renderings are first generated with pre-defined and randomly selected camera angles and are further inferenced by the material segmentation model and attached to a stacked temporary UV map. Weighted voting and region unification are further applied to generate the final material UV.**

alignment using the text-image Contrastive Language-Image Pre-training(CLIP) [32, 51, 57] model. DreamFusion [55] replaced CLIP with a diffusion model and introduced the loss of Score Distillation Sampling (SDS) to extract knowledge from the denoising process. Magic3D [37] further enhanced generative performance by adopting a coarse-to-fine framework and employing grids as a 3D representation in the second stage. Additionally, some other methods [1, 10, 11, 27, 52, 53, 60, 75] have combined NeRF techniques with the diffusion-based text-to-image models, proposing NeRF-based generators, but they primarily focused on geometric generation and often overlooking appearance.

## 2.2 Surface Material Generation

Generating realistic PBR material information such as metallic and roughness on the surface of 3D assets is key to making the asset look like a real object, that will dictate how surfaces interact with incident light, determining asset surface reflective behavior and color variations.

Traditional material generation methods predominantly focus on predicting physics-based materials under given lighting conditions, often requiring intricate multi-view [4] or polarizing [21] equipment. These methods often use synthetic data to train single-view Spatially Varying Bidirectional Reflectance Distribution Function (SVBRDF) prediction networks [19], which are then combined with other single-view data [25, 47] or custom training strategies [20, 36, 67] to obtain predicted material textures. These methods generated surface material information that looks inconsistent with what we perceive in the real world.

In recent years, many work have appeared in the field of 2D material segmentation for the controllable generation of materials in the form of SVBRDF maps [34, 58, 66, 68]. Based on a similar idea, there are several new work have also emerged in the field of 3D material generation in an attempt to estimate materials under natural light conditions, Fanasia3D [13] decouples geometric and appearance modeling, using Bidirectional Reflectance Distribution Function (BRDF) to generate photo-realistic textures. However, it always predicts materials entangled with environmental lights, which leads to unrealistic renderings under novel lighting conditions. PhotoScene [78] utilizes procedural graphs as a prior for materials, generating high-resolution tiled material textures for each object in a scene, along with globally consistent lighting for the entire scene. PhotoScene, DiffMat [79], and Material Palette [43] are tailored for

tiled material generation. However, the surface material of a single complex 3D asset is often not tiled, making it difficult to generate and represent the asset's true appearance through simple tiling. MatAtlas [9] generates relightable textures for 3D models given a text prompt with GPT4-V, but its generations across different views might differ in the appearance of the details.

## 2.3 Existing 3D and 2D Datasets

When considering learning prior information about the surface material, the first step is collecting enough data to support running the training process. In recent years, there have been some large-scale 3D datasets released, one of the most representative is the Objaverse, which is divided into Objaverse-1.0 [18] and Objaverse-XL [17], with approximately 800,000 and 10 million 3D assets, respectively. However, 3D assets in Objaverse generally lack material information, posing a limitation for research on surface appearance generation. Other 3D datasets like KIT [33], YCB [8], BigBIRD [59], and pix3d [63] offer calibrated models for various household objects, but they suffer from a severe lack of scale, containing at most a few hundred objects. Larger photorealistic object datasets [23, 54] and CAD model datasets [35, 70, 72] all do not include Albedo or material information. These existing 3D datasets fail to meet the requirements for generating realistic surface materials UV maps for individual complex 3D assets.

Due to the relative ease of obtaining 2D images, 2D material segmentation has accumulated more extensive large-scale datasets than 3D in the past decades. Such as the DMS dataset [65], encompassing 44,560 indoor and outdoor images with annotations for 3.2 million dense segments. The OpenSurfaces dataset [6] contains annotations for 37 material categories on 19,000 images of residential indoor surfaces. MINC [7] hosts the largest texture recognition dataset, featuring 3 million points annotated for 23 materials across 437,000 images. While these 2D datasets are extensive, their labels are often tailored for multi-object scenarios, bringing too much training noise when learning 2D priors for single-object scenarios.

## 3 SIGNIFICANCE OF MATERIAL

Creating high-quality materials in computer graphics is a challenging and time-consuming task, which requires great expertise. 3D assets with the correct materials can present the same impressions as in the real world under various lighting conditions. Components of the asset with different PBR materials will result in various reflection

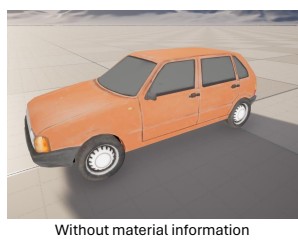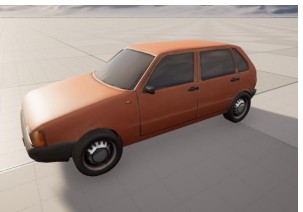

**Figure 3: Comparison of 3D assets rendered with and without PBR material information under the same lighting conditions.**

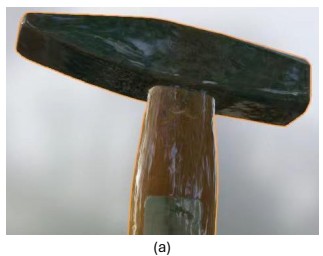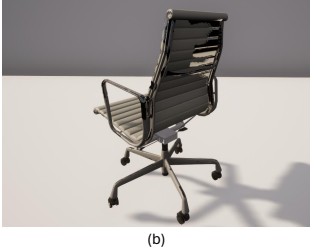

(a)                                        (b)

**Figure 4: Case analysis of AI-generated asset surface material. (a) shows the rendering effect with the PBR material set to a fixed value on different structural components. (b) shows the generated material information cannot be consistent within the same semantic area.**

**Table 1: Statistics on the frequency of occurrence of different material categories contained in each image.**

| Material label | Number | Material label | Number |
|---|---|---|---|
| metal | 935 | brick | 186 |
| wood | 842 | porcelain | 163 |
| plastic | 768 | clay terracotta | 154 |
| glass | 712 | concrete | 152 |
| paint | 626 | nylon | 75 |
| rubber | 524 | rusty metal | 53 |
| leather | 437 | ston | 46 |
| fabric | 391 | bone | 25 |
| fruit&leaf | 273 | bamboo | 22 |
| flower | 252 | others | 181 |

effects even under the same illumination. 3D assets without PBR material information will cause extreme distortion when rendering diversified illuminations, making these properties inapplicable for real-world demands. Visualization can be found in Fig.3.

After recognizing the importance of PBR materials for 3D assets, we have conducted our early attempts to explore the potential of existing public datasets of 3D assets. In the newly proposed large-scale 3D object datasets Objaverse [17], we have analyzed a total of more than 270,000 assets of various categories, while only about 3k assets are attached with realistic PBR material information. This lack of material information in Objaverse makes it hard to learn the distributions of material semantics from the provided 3D assets.

Although there are some of the latest 3D asset generation methods [9, 13] claimed to have provided surface materials for the AI-generated content, the surface material quality of the 3D assets generated by these methods is rather poor with obvious distortions [58, 68], mainly caused by the following two problems. One of the problems that happened in some methods is that the PBR materials (Metallic, Roughness, *etc.*) attached to the surface are pre-defined fixed values regardless of the Albedo or semantic information. As shown in Fig. 4(a), the same PBR material values are attached to the handle and the head of the hammer, but they should be two materials with significant differences. Another issue is that the generation of the PBR material lacks guidance from real-world common sense or prior knowledge. The materials attached to a continual region of the asset may be discontinuous or unconvincingly related to the actual semantics of that region. A case is shown in Fig. 4(b), the region of the back of the chair should be applied with a continual material such as *fabric* or *nylon*, but *metal* is mistakenly attached in some part of the region.

Inspired by such case studies, we consider that human prior knowledge of witnessed categories of 3D assets can be utilized to judge or supervise the generation of the surface material. This statement also explains the logic of modelers manually assigning PBR materials to different assets, making it more convincing and logical. Further, we surveyed 100 people about the materials they thought were likely to appear in different categories of objects and showed each person 10 pictures of indoor and outdoor scenes. Each person was asked to count what materials might occur in every image, and the results are shown in Tab. 1. The survey results ensure that humans can confidently infer material information from a 2D image, and the frequency of different materials that occur in common objects is also supposed to be determined. This result greatly supports our motivation to introduce prior 2D knowledge to surface material generation of 3D assets.

## 4 MIO DATASET

### 4.1 Motivation for Establishment

Our pilot research indicates that 2D prior knowledge from humans can provide strong guidance and supervision for generating surface material on 3D assets. The following questions will be about how to employ and where to obtain such material prior knowledge related to 3D asset generation. Inspired by our relevant knowledge of the computer vision area, we figured out that perception-based methods can intuitively learn prior knowledge from training data into the models and infer the samples accordingly. Considering providing dense surface PBR material on 3D assets, segmentation is the most suitable method as it can provide pixel-wise prediction of material classes.

As aforementioned, in early attempts, we tried to collect available material information from public 3D asset datasets to build prior knowledge but ended up due to the extreme lack of material information. We subsequently notice that compared with 3D assets, 2D images are easily accessible through public websites or datasets with much wider distribution and total amounts. However, domain gaps exist between the distributions of 3D asset multi-view renderings and the existing annotated 2D image datasets, which makes learning & applying material prior knowledge less accessible. Therefore, we

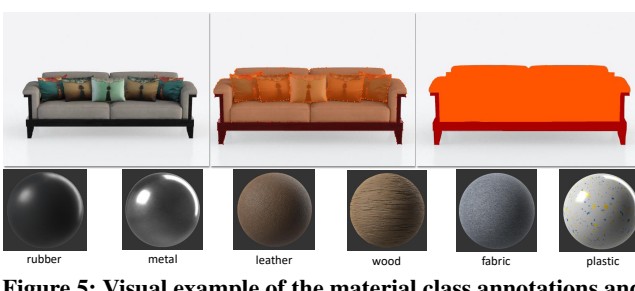

**Figure 5: Visual example of the material class annotations and the mapping with PBR material spheres.**

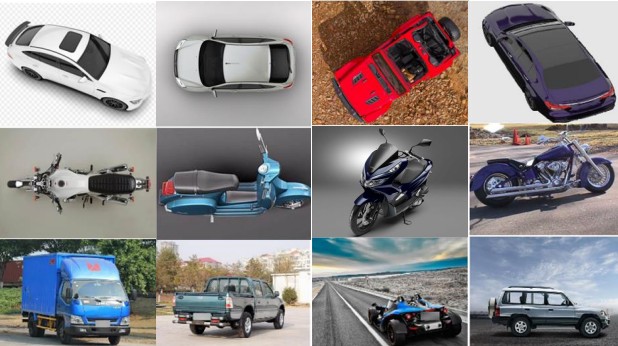

**Figure 6: Visual displays of samples in metaclass *Cars* collected in MIO dataset.**

**Table 2: Statistics of material labels occurrence in images.**

| Material label | Number | Material label | Number |
| --- | --- | --- | --- |
| metal | 8,946 | fabric | 3,373 |
| wood | 7,088 | fruit&leaf | 1,742 |
| plastic | 6,928 | flower | 1,677 |
| glass | 5,802 | brick | 1,017 |
| paint | 5,626 | porcelain | 921 |
| rubber | 5,324 | clay terracotta | 910 |
| leather | 3,417 | concrete | 794 |

**Table 3: Number of rendered images and real images of every metaclass in the MIO dataset.**

| Class name (*abbr.*) | Rendered image | Real image | Total image |
| --- | --- | --- | --- |
| Furniture (*fur.*) | 4,152 | 5,455 | 9,607 |
| Cars (*car*) | 1,935 | 4,117 | 6,052 |
| Buildings (*bui.*) | 418 | 1,752 | 2,170 |
| Musical Instrument (*ins.*) | 627 | 1,637 | 2,264 |
| Plants (*pla.*) | 552 | 2,417 | 2,969 |

were motivated to construct a customized 2D image dataset that perfectly fits the demand of providing robust prior knowledge for surface material.

## 4.2 Data Collection and Annotation

To overcome possible domain gaps between 2D images and 3D asset renderings, we tried to collect and construct the image samples of our dataset under the following guidelines: **(a)** Each image sample could only contain one out-standing foreground object; **(b)** Image samples should be collected with similar amounts from both real-world scenes or renderings of 3D assets; **(c)** Image samples should be captured from diverse camera angles, including some especial angles such as the top view or bottom-side view. With the above guidelines, we ensure the gathered images share similar distributions with multi-view renderings of 3D assets, which largely guarantees the accuracy of further material predictions.

The sources of the collected images are freely accessible public datasets [31, 77] and 2D image renderings from 3D objects in website photo libraries [5]. In addition, we also procured some well-designed 3D assets that are used for game development and expanded the data collection by rendering multi-view images of these high-quality assets.

The biggest difference between our dataset and existing 2D segmentation datasets is that our customized dataset is designed to build extra alignments between semantic labels of different material classes and real PBR material values (Metallic, Roughness) for the included materials. The accuracy of the image annotation affects the overall performance of the material segmentation model trained

on the dataset, while the authority and rationality of the mapping between material class annotations and PBR materials influence the final rendered visualization of the assets. The number of material categories included in the dataset and their mapping relationships with PBR materials were discussed and determined by a group of nine professional 3D asset modelers. They have drawn upon their modeling expertise and considered the survey results shown in Tab. 1 to collect PBR material sphere candidates from more than 1,000 real PBR material spheres from public material libraries such as ACG [3] or Adobe Substance 3D Painter. Finally, 14 material categories, together with the mapping with the PBR materials to be the label space of our dataset.

After confirming the number of material categories in the dataset, we cooperated with a large and highly specialized annotators team to conduct pixel-wise dense annotations on the collected image samples. Based on the design of the dataset, we require that only the foreground objects contained in each picture be labeled with materials, and the background part is set to the background class regardless of the semantics. Each image sample was first annotated through an application driven by Segment Anything [34] and manual refinements, and sent to other annotators for multi-round re-annotation. Each annotator can handle approximately 50 images per day, ensuring the quality of the annotations is precise and accurate. Fig. 5 illustrates the annotations and alignments of material information in one of our samples.

## 4.3 Dataset Distribution

The dataset is named **Materialized Individual Objects (MIO)**, containing single-object image samples captured under diverse camera poses and annotated with convincing material labels and PBR material values. The MIO dataset comprises 23,062 multi-view images of individual complex objects, annotated into 14 material classes and categorized into five metaclasses: furniture, cars, buildings, musical instruments, and plants. The occurrence of each label is shown in Tab. 2. Occurrence frequency statistics of each metaclass belonging

to real images and asset-rendered images are illustrated in Tab. 3. Approximately 4,000 top-view images are included in the MIO dataset, providing a unique perspective rarely found in existing 2D datasets. Some image samples with the metaclass *cars* are displayed representing the diversity of camera poses and distributions, shown in Fig. 6.

# 5 METHOD

## 5.1 Material Segmentation

Inspired by existing semantic segmentation methods trained under material semantic labels, we establish a material segmentation process that better fits the demands of 3D assets. Compared with current semantic datasets annotated with material information, material segmentation focuses on dense predictions of a single object under diverse poses and camera angles. Given an image $I$ with pixel-wise RGB value $x$ and annotated material label $y$ as a pair $< x_i, y_i >$ for each pixel $i$, the material segmentation network encodes visual features from the input image and decodes the features into per-pixel possibility vectors $P_i = (p^{(i,1)}, p^{(i,2)}, ..., p^{(i,n)})$ for $n$ different classes at pixel $i$. The final prediction of each pixel can be calculated from $P_i$ through *argmax* function.

We notice that the Segment Anything Model (SAM) [34] has shown its ability to handle semantic region segmentation on single-object images in previous work [71]. Thus, we formulate the material segmentation network with a modified ViT [22] backbone using pre-trained segmentation weights from SAM-b model. The decode head follows the setting in UperNet [73] with cross-entropy loss as supervision. To prevent possible long-tail problems caused by imbalanced training data, we adopt a class-balanced sampling strategy [16] to enhance the robustness and generalization ability of the model. During training stage, the cross-entropy loss can be calculated with:

$$L = -\frac{1}{HW} \sum_{i=1}^{HW} \sum_{c=0}^{n-1} y^{(i,c)} log(p^{(i,c)}), \qquad (1)$$

where $H, W$ denotes the shape of the input image, $n$ denotes the number of the classes, $y^{(i,c)}, p^{(i,c)}$ represents the ground truth value, and the predicted possibility of class $c$ at pixel $i$.

## 5.2 MaterialSeg3D

In this section, we introduce a novel material generation method, named **MateriaSeg3D**, a workflow that generates precise material information for 3D assets. The proposed MateriaSeg3D includes three components: multi-view rendering, material prediction, and material UV generation, as shown in Fig. 2. Specifically, in the multi-view rending stage, the workflow first defines diverse camera poses capturing 360° of the target assets. 2D rendering images can be obtained from various angles from specific camera poses. In the material prediction stage, the material segmentation model is trained beforehand and infers multi-view renderings captured in the previous stage into the predicted material labels. In the material UV generation stage, predicted results of the renderings are first projected back to temporary UV maps and are further processed through a weighted-voting mechanism to obtain the final material label UV. Pixel values of the material label UV can be further transformed into PBR material (Metallic, Roughness) with the mapping relationships

between labels and material spheres. We will introduce the details in the following subsections.

**Multi-View Rendering.** In order to provide dense material predictions on the entire surface of an object, the elevation and rotation matrices of the rendering camera should cover 360° of the entire asset. Therefore, we first manually define five specific camera angles with the elevation and rotation status at $(90°, 0°)$, $(15°, 0°)$, $(15°, 90°)$, $(15°, 180°)$, $(15°, 270°)$. These rendered views can provide high-quality results and serve as popular views for human inspection. Next, we equally divide the entire 360° rotation into 12 different directions, on which there will be three different elevation angles, 0° as a fixed value, and the other two will be randomly selected within the range of $(0°, ±30°)$ respectively. Through this, the renderings can provide visual information about all surfaces of the object, including the top and bottom. The manually selected views will further present additional constraints during the ensemble stage of the material UV.

**Material Prediction.** Following the details presented in Sec. 5.1, we can obtain a material segmentation model capable of predicting accurate material labels on images captured from various views. This model is used to infer the material information of the multi-view renderings of the input object. The predicted material labels are then used to generate material UVs.

**Material UV Generation.** After acquiring the predicted material results on the multi-view renderings, we generate the PBR material UV map for the 3D asset by attaching the material information to the pixel-wise UV map. Specifically, for each rendering with the rotation and elevation angle, we assign the predicted material labels to the corresponding pixel coordinates in the Albedo UV and form a new temporary material label UV. Through this, we can obtain a group of single-angle material label UV maps $M_{view} = M_1, ..., M_n$, where $n$ represents the number of the sampled camera views mentioned in the earlier paragraph.

As each rendering view can only provide limited material label information on the entire UV map, instead of sequentially updating the material label UV [12], we introduce a weighted voting method to decide the final material label of each pixel on the UV map. As aforementioned, five manually selected views will have higher weights when voting. Thus, the voted material label UV map can be calculated as follows:

$$M_{material} = vote(\alpha(M_1, M_2, M_3, M_4, M_5), M_6, ..., M_n), \qquad (2)$$

where $\alpha$ denotes the weighting factor of the high-value views, and we set $\alpha = 2$ in our experiments.

While the pixel values of the material label UV map are class labels predicted from the material segmentation model, the PBR material (Metallic, Roughness) UV map used to render visual effects can be transformed from the mapping relations between class labels and material spheres defined in the dataset.

# 6 EXPERIMENTS

## 6.1 Implementations & Evaluations

Learning precise 2D material prior information is at the forefront of our MaterialSeg3D pipeline for raw 3D objects. We trained our model with SAM-b[34] pre-trained ViT [22] backbone. The optimizer is AdamW [44] with the learning rate and weight decay are

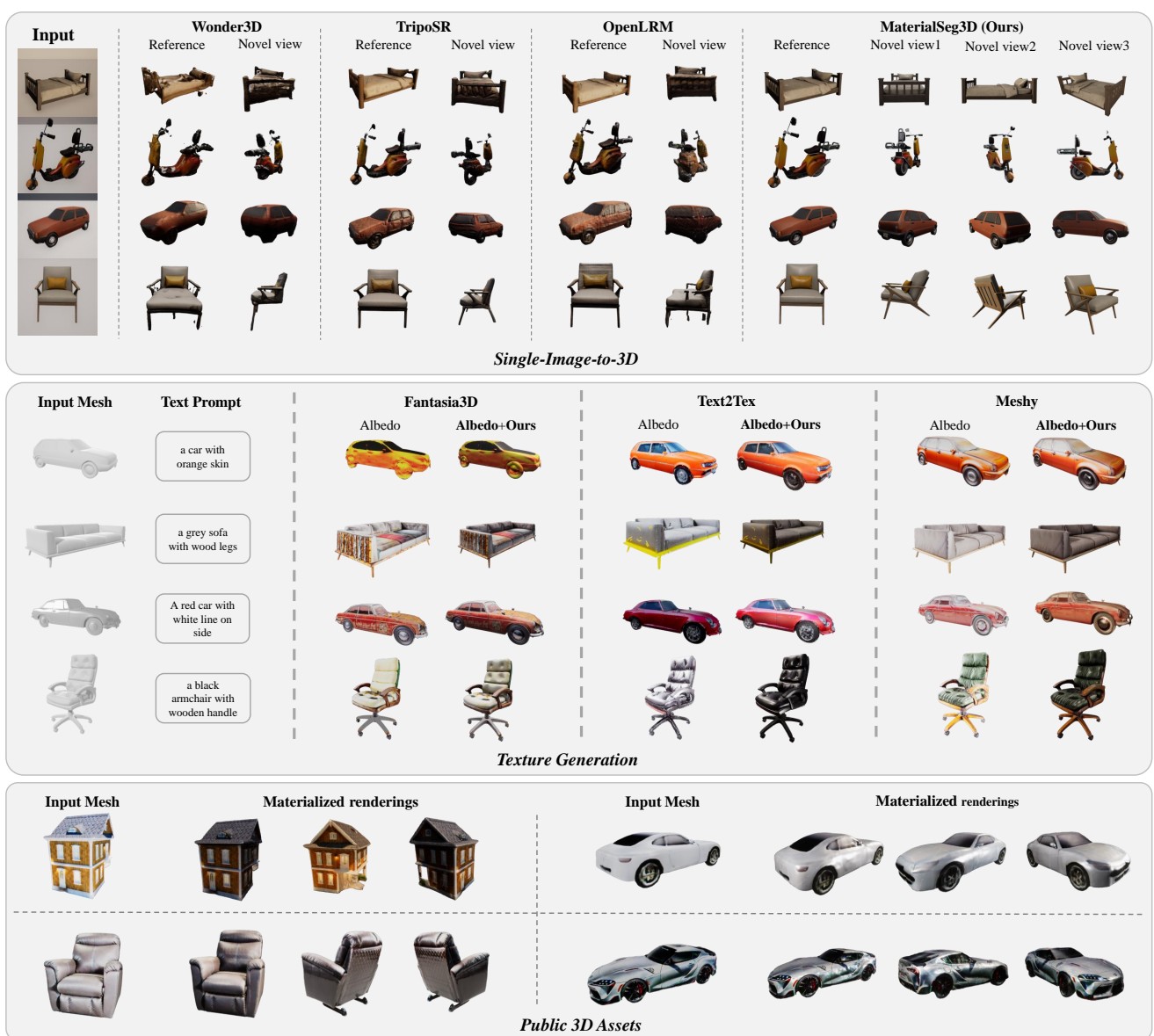

**Figure 7: Detailed visual comparisons between MaterialSeg3D and previous method from three aspects: single-image-to-3D generation methods, texture generation methods, and public 3D assets.**

$6 \times 10^{-5}$ and $1 \times 10^{-2}$, respectively. We set batch size = 8 and training iterations = $80k$, and images are resized to $1024 \times 1024$. All experiments are conducted under MMsegmentation [15] framework and on 4 **80G NVIDIA A100 GPUs**.

## 6.2 Compared with Previous Work

**Material Segmentation.** To evaluate the effectiveness of our proposed material segmentation method mentioned in Sec. 5.1, we apply five widely-used and state-of-the-art semantic segmentation backbones as comparisons to train segmentation models on the MIO dataset. We provide mIOU performance comparisons between these

methods on Objaverse [18] samples and the test image set of the MIO dataset. We randomly sample 50 assets with ground-truth PBR material UV from Objaverse and evaluate the accuracy of the output material label UV from MaterialSeg3D with the ground-truth. Quantitative results are shown in Tab. 4. It can be observed that our material segmentation method outperforms all the other semantic segmentation backbones, providing accurate and reliable material predictions for further renderings.

**Overall Performances.** To evaluate the effectiveness of the proposed material generation method, we compare previous approaches from the following three aspects: single-image-to-3D generation methods,

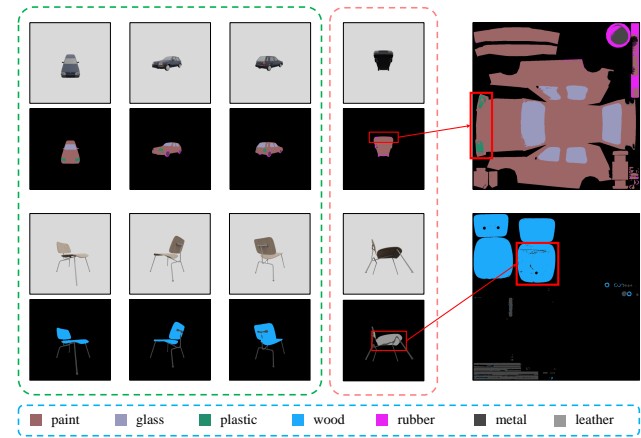

| paint | glass | plastic | wood | rubber | metal | leather |

**Figure 8: Visualization of the segmentation results on multi-view rendering of 3D assets and the colored material UV map acquired from the weighted voting mechanism.**

**Table 4: Quantitative results about the performance of the semantic segmentation methods on the test split of MIO dataset / material label UV of Objaverse.**

| Method | MIO Dataset (%) | | | | | | Objaverse Dataset (%) | | | | | |
|---|---|---|---|---|---|---|---|---|---|---|---|---|
| | car | fur. | bui. | ins. | pla. | mIOU | car | fur. | bui. | ins. | pla. | mIOU |
| ConvNeXt[41] | 71.03 | 74.85 | 69.33 | 72.40 | 76.72 | 72.87 | 75.35 | 76.04 | 72.34 | 76.72 | 78.95 | 75.88 |
| HRNet[62] | 75.71 | 79.94 | 76.37 | 80.14 | 81.35 | 78.70 | 78.40 | 78.83 | 76.03 | 82.00 | 81.40 | 79.33 |
| ViT[22] | 73.96 | 77.67 | 75.53 | 79.45 | 78.66 | 77.05 | 77.33 | 78.45 | 75.70 | 81.38 | 78.36 | 78.24 |
| Swin-T[40] | 75.09 | 79.04 | 78.45 | 80.92 | 81.40 | 78.98 | 78.89 | 79.77 | 78.64 | 82.97 | 82.01 | 80.46 |
| MAE[28] | 76.42 | 82.06 | 77.59 | 82.74 | 85.92 | 80.95 | 79.61 | 81.28 | 76.96 | 83.41 | 86.37 | 81.53 |
| Ours | **81.83** | **85.22** | **81.76** | **84.39** | 86.38 | **83.92** | 82.75 | 84.33 | 81.14 | 84.33 | 87.76 | 84.06 |

texture generation methods, and public 3D assets. The corresponding results are shown in Fig. 7. Considering single image-to-3D generation methods, we compare state-of-the-art Wonder3D [42], TripoSR [64], and OpenLRM [29] in this section. Specifically, given a reference view as input, Wonder3D, TripoSR, and OpenLRM generate a 3D object with referenced texture. We can observe that the provided MaterialSeg3D significantly outperforms the previous work owing to the adoption of well-defined 3D mesh and Albedo information. Fairly comparison, we modify existing texture generation methods like Fantasia3D [13], Text2Tex [12], and online functions provided by *Meshy* [1] for evaluation. Given a well-defined geometry mesh, previous work provide texturing results according to the text prompt as shown in Fig. 7. The results demonstrate our method provides much more realistic renderings under different lighting conditions. Note that for Fantasia3D, we only adopt its texture generation (Appearance Modeling) stage during comparison. Moreover, we also provide material generation results for 3D assets obtained from public websites, exampling as *tripo3d* [2] and *turbosquid* [3]. From the results in Fig. 7, we can observe the proposed MaterialSeg3D can generate precise PBR material information while significantly improving the overall quality of the assets.

Furthermore, we also provide quantitative results comparing our method and existing Image-to-3D methods including Wonder3D [42], TripoSR [64] and OpenLRM [29]. We adopt CLIP

[1] https://app.meshy.ai/
[2] https://www.tripo3d.ai/app/
[3] https://www.turbosquid.com/

**Table 5: Quantitative evaluations from reference view and novel views on samples from Objaverse-1.0 dataset.**

| Method | Evaluation view / mesh | CLIP Similarity↑ | | PSNR↑ | | SSIM↑ | |
|---|---|---|---|---|---|---|---|
| | | Reference | Novel | Reference | Novel | Reference | Novel |
| Wonder3D [42] | w/o | 0.85 | 0.84 | 16.06 | 15.83 | 0.78 | 0.75 |
| TripoSR [64] | | 0.93 | 0.90 | 16.93 | 16.14 | 0.79 | 0.76 |
| OpenLRM [29] | | 0.92 | 0.87 | 16.30 | 15.37 | 0.77 | 0.76 |
| Baseline | w | 0.93 | 0.93 | 16.28 | 16.30 | 0.79 | 0.78 |
| Baseline + Ours | | **0.98** | **0.97** | **20.72** | **18.39** | **0.85** | **0.84** |

Similarity [57], PSNR, and SSIM as the evaluations, and the corresponding results are shown in Table 5. We choose assets from Objaverse-1.0 dataset [18] as the test sample and randomly select three camera angles as novel views. The ground-truth reference and novel views are captured from assets with ground-truth material information and fixed lighting conditions. Given a well-defined 3D mesh and Albedo, our workflow can provide reliable PBR material, resulting in more realistic rendering visual effects.

## 6.3 Visualization on Weighted Voting

To illustrate the effectiveness of the weighted voting mechanism in the material UV generation stage, we provided visualizations of multi-view material segmentation results and the final material label UV maps, shown in Fig. 8. Although some regions might be predicted as wrong materials in some tricky angles, the correct predictions of the same region from other views will correct the final material labels through the weighted voting mechanism of the temporary UV maps.

## 7 LIMITATION

One of the limitations of our work is that current 3D asset generation methods mostly bake specific illuminations onto the generated RGB textures. Applying our workflow to the Albedo UV coupled with light reflections will lead to unrealistic visual effects under different illuminations.

Another limitation is that the quality of the input mesh will largely influence the generation of surface material and visual renderings. When applying our workflow on low-quality coarse meshes with uneven surfaces, the results are less satisfying. Detailed explanations and visualizations can be found in Supplementary Materials.

## 8 CONCLUSIONS

In this paper, we innovatively introduce the idea of adopting 2D prior knowledge of surface material in the material generation of 3D assets. We propose **MaterialSeg3D**, a novel workflow that takes a geometry mesh and Albedo UV as input, and generates dense PBR material information with the supervision of 2D prior knowledge. We also establish a 2D single-object material segmentation dataset **MIO** including images collected from diverse distributions and camera poses, thus providing strong 2D prior knowledge for the material segmentation model. Extensive experiments show the effectiveness of our proposed workflow. The workflow and the dataset show its ability to complete missing PBR material information for the public 3D assets, providing convenience for subsequent studies.

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
