# OpenReview forum: "MaterialSeg3D: Segmenting Dense Materials from 2D Priors for 3D Assets"
_acmmm.org/ACMMM/2024/Conference — MM2024 Oral_

### Official Review · Reviewer_N7Xu · 2024-05-05

**Rating:** 4
**Confidence:** 2

**Summary:**

The authors proposed to incorporate material segmentation priors from multi-view images to illuminate 3D assets, inspired by humans' ability to deduce an object's material based on its appearance and semantics. They introduced a dataset called MIO to generate these material segmentation priors and conducted experiments on MIO and Objaverse to assess the method's effectiveness. The results showed that the proposed method produced more vivid illuminations.

**Strengths:**

1. The authors have proposed a new dataset that appears to be distinct from existing datasets, which has the potential to facilitate future work on this topic.
2. The motivation behind leveraging material segmentation priors is intuitive, and the entire paper is clear and easy to understand.
3. The experimental results and visualization are good.

**Limitations:**

1. The reviewer has raised concerns about the fairness of the experiment presented in Table 4. The proposed method is fine-tuned on the SAM-b pre-trained ViT backbone. However, it is not clear what the pre-trained weight is for the other methods.
2. The experiments were conducted using 4 80GB NVIDIA A100 GPUs, indicating a high computational load. The reviewer is curious about the efficiency of the proposed method compared to prior works.
3. Figure 2 is located on Page 3, while it is referenced in Section 5 on Page 6, making it difficult for readers to quickly access the information.
4. There appear to be typographical errors at the bottom left of Page 1.

**Suitability:**

3

---

### Official Review · Reviewer_qZEn · 2024-05-21

**Rating:** 5
**Confidence:** 2

**Summary:**

To tackle the problem of material assignment for 3D asset creation, this paper collects MIO, a 2D dataset, where the material is annotated based on human prior knowledge of 2D images. Furthermore, the paper proposes the MaterialSeg3D workflow for material segmentation and material UV prediction.

**Strengths:**

- The motivation behind this paper is strong: PBR maps will improve the rendering effects and human priors can provide extra insights in predicting materials, as proved by Fig 3 and Tab 1. The writing is clear and the story is well-explained.
- The paper presents solid results, e.g., it has an outstanding advantage over prior methods in segmentation.

**Limitations:**

- The authors are claiming the MIO dataset as a contribution, will it be released?
- It is mentioned that multi-view images are needed in the workflow and the MIO dataset contains multi-view images. However it’s unclear how such data is collected from the 2D datasets. Are they collected from videos? More specifications should be provided.
- The distribution of classes in the MIO dataset is very unbalanced, as most examples fall in the category of furniture and cars. Will this have any bias in material prediction?
- For material prediction, the current training does not guarantee the diversity of the assigned labels. But in the real scenario, the same object often has different combinations of materials (e.g., a chair can be plastic or metal). How to improve the model to have more diversity?
- In Sec 5.2, the material UV is decided by a voting mechanism. If a small aggregation network is used here on the 5 views, will it have better performance?
- In the bottom-right of Fig 7, the generated texture on the car does not look very smooth and exhibits white blobs. Are these artifacts?
- It would be helpful to add a user study to demonstrate the effectiveness of the proposed method.

**Suitability:**

3

---

### Official Review · Reviewer_Us99 · 2024-05-26

**Rating:** 5
**Confidence:** 2

**Summary:**

Recent 3D object generation works bake the illumination and shadow into the texture, thus are not able to relight the generated object and limit the downstream applications. The paper proposes MaterialSeg3D, a 3D asset material generation framework to solve the above mentioned problem. It utilizes 2D material segmentation as a prior, by projecting 3D assets into different views and applying 2D material segmentation method, multiple material UV map of different views will be generated. Then a weighted-voting mechanism will be used to obtain the final material UV map of the 3D asset. Besides, a 2D material semantic segmentation dataset MIO is proposed to further benefit the community.

**Strengths:**

- A single asset 2D material semantic segmentation dataset is proposed, with precise and accurate human annotations
- A 2D material semantic segmentation model is proposed, and outperforms existing state-of-the-art methods.
- MaterialSeg3D only relys on a pre-trained 2D segmentation model, no any 3D data or network is needed.
- MaterialSeg3D outperforms state-of-the-art 3D generation methods and have a much better visual quality.
- Extensive experiments are provided to demonstrate the performance of both 2D material segmentation and 3D material generation of the proposed method.

**Limitations:**

- Just as mentioned in the limitation part, current 3D generation approaches will bake illumination inside the texture, directly apply the proposed MaterialSeg3D will lead to unrealistic visual effects.
- The paper doesn’t mention how the results in Figure 7. (Single-Image-to-3D) are generated. Since MaterialSeg3D is a material generation framework, how do you get the 3D asset?
- In table 4, some columns highlight the best method, some columns don’t. It would be better to make them consistent.

**Suitability:**

3

---

### Official Review · Reviewer_xPb7 · 2024-05-26

**Rating:** 5
**Confidence:** 3

**Summary:**

This paper presents a novel 3D asset generation method that adopts 2D prior knowledge of surface material for achieving photorealistic visual effect.  Multi-view renderings of a 3D asset are generated and processed using a 2D material segmentation model pretrained on the MIO dataset. Then, the segmented material maps from different views are merged into a final UV map using a weighted voting scheme. The final UV map is converted into a Physically-Based Rendering (PBR) material map, incorporating properties like metallic and roughness to ensure realistic interactions with light. This framework promises to streamline 3D asset creation and enhance the realism of virtual environments.

**Strengths:**

•	This paper is well-written and easy to follow.

•	Their proposed method looks quite innovative and it addresses a significant gap in current 3D asset generation techniques, where materials often lack the necessary detail and accuracy for realistic rendering.

•	MIO dataset is a substantial contribution to the field, providing a foundation for future research and practical applications.

•	The visualizations provide clear and insightful comparisons with other methods.

**Limitations:**

Overall, I think this paper makes significant strides in improving 3D asset creation and material generation. Besides, I have some concerns on its generalization. How to ensure that the models generalize well to a diverse range of objects and materials? How is the light condition encoded?

**Suitability:**

3

---

### Meta-Review · Area_Chair_tJd8 · 2024-07-02

**Recommendation:** Accept (Oral)
**Confidence:** 4

**Metareview:**

The article is focused on 3D asset creation and material generation. Unanimously among the 4 reviewers, it is well-written, the proposal is clearly evaluated as sufficiently innovative by proposing a significant gap in current 3D asset generation techniques. The responses of the authors have solved most of the few concerns of the reviewers.